# SELF-TAILORING PROMPTS FOR PARAMETER EFFICIENT TUNING SPEECH RECOGNITION

## ABSTRACT

Soft-prompt tuning is an emerging topic for speech recognition despite its success in many natural language processing tasks. Although it appears to be a promising approach for efficiently fine-tuning large speech models, it can suffer from subpar prompt generalization and a lack of instance-specific guidance due to its "one-size-fits-all" template. To address these limitations, we propose a self-tailoring prompting mechanism that adaptively modifies the prompt tokens to incorporate relevant speech utterance-specific information. Self-tailoring mechanism includes simple yet effective prompt masking regularization techniques and a redundancy reduction loss to improve the quality of soft prompt tokens. Extensive experiments demonstrate that our method achieves better generalization capability and consistently achieves improved performance on the speech recognition task under a wide range of acoustic scenarios, including both clean and noisy speech environments. Self-tailoring prompt tuning outperforms the full fine-tuning model with as few as 0.7% of its trainable weights.

## 1 INTRODUCTION

In recent times, self-supervised pre-trained speech models have emerged as transformative and indispensable technologies for advancing speech processing. These sophisticated systems (Baevski et al., 2020; Hsu et al., 2021; Huang et al., 2021; Ng et al., 2023c; Chen et al., 2022; Radford et al., 2023; Baevski et al., 2023) reduce the heavy reliance on costly and laborious hand-transcribed speech data by leveraging the power of readily available, unannotated audio data to acquire rich and meaningful representations of spoken language. This potent representation learning has expanded the horizons of possibilities in various speech-related tasks (Ni et al., 2023; Gong et al., 2023; Jiang et al., 2023), especially in low-resource data settings. Nevertheless, despite the remarkable strides made by this approach, it remains crucial to fine-tune these large models for practical use. Training the model on all downstream tasks is computationally expensive and lacks scalability, as it necessitates the training of millions of parameters for each task (Chen et al., 2023).

Soft-prompt tuning (PT) (Li & Liang, 2021; Lester et al., 2021; Kim et al., 2023) is an alternative approach to full model fine-tuning that falls under the category of Parameter-Efficient Tuning (PET) methods. It is designed to overcome the aforementioned limitations by tuning only a specific set of trainable parameters linked to a fixed number of pre-defined latent tokens known as '*prompts*'. These prompts are prepended to the input embeddings, effectively serving as guiding signals that enhance the model's optimization of the downstream task at hand. The trainable prompts are updated to improve their ability to refine the latent representations, while the pre-trained model remains entirely frozen and untouched during the fine-tuning process. For tasks like intent classification, speaker diarization, and speech generation (Chang et al., 2023b; Guo et al., 2023), prompt tuning has been demonstrated to be effective, or even outperform full-model fine-tuning of the large model.

However, it has become evident that the efficacy of soft prompts is predominantly influenced by the scale and diversity of labeled training data (Su et al., 2022). Several studies have unveiled the limitations of prompt tuning, particularly in various few-shot classification tasks (Gu et al., 2022a; Guo et al., 2022), and its challenges in facilitating robust domain adaptation (Goswami et al., 2023), especially in low-resource settings. Similarly, we applied prompt tuning to speech recognition and found that it was less beneficial for changes in speaker and noise type distortions when the utterance distribution deviated from the training set. Understandably, once training is completed and prepared

for deployment, soft-prompts become a standardized template conditional on the training speakers within the limited noisy environments for all inference samples. In practice, these soft prompts may suffer from subpar prompt generalization due to limited training data diversity and occasionally serve as a sub-optimal "***blanket solution***", lacking the precision needed for instance-specific guidance.

In this paper, we introduce a self-tailoring prompting mechanism to improve the effectiveness of soft prompt tokens for speech recognition systems. This mechanism adaptively modifies the prompt tokens to incorporate relevant utterance-specific information from each individual sample, enabling a more versatile format of the soft prompt tokens that target better focus fine-tuning of speech representations and improve performance stability and generalization. Central to this mechanism is an external lightweight attribute encoder that extracts ASR task-relevant global attributes of the acoustic timbral features at the utterance level and seamlessly integrates them with the soft prompts to generate utterance-level prompts, as illustrated in Figure 1. This approach aligns with the overarching goal of preserving optimal performance while minimizing the number of trainable parameters, preserving the principles of PET that underlie the efficacy of our method. Besides, we also propose and demonstrate useful soft prompt regularization techniques and a regularization auxiliary loss function, which, to the best of our knowledge, have not been previously applied to enhance the quality of soft prompt tokens. In summary, our primary contributions include:

1. **We propose a highly effective self-tailoring mechanism for soft prompts** that enables significant adaptability in the parameter efficient tuning speech recognition, thereby addressing the limitations associated with conventional prompt tuning methods and their tendency to rely on a *one-size-fits-all* approach. We analytically validate the proposed method's ability to generalize across different existing pre-trained models.

2. **We further improve the robustness and informativeness of prompts** with the proposed *prompt masking* regularization strategy and *redundancy reduction loss* for soft prompt tokens. These techniques serve to maximize the functionality and efficiency of the prompts, enabling the prompts to offer more valuable and unique guidance during both training and inference.

3. **We conduct extensive experiments on a wide range of acoustic scenarios (e.g., clean and real-world noisy speech)** to evaluate the generalizability performance of our method. We consistently achieve a relative gain of 5% in most cases, demonstrating that our framework is highly robust and domain-agnostic. It prominently outperforms the full-finetuning model with self-tailoring prompt gating, which utilizes only 0.7% of its trainable weights.

## 1.1 RELATED WORK

**Learning based strategies** To improve the prompts generalization capabilities, previous research has achieved better prompting performance by employing larger pre-trained models and working with extensive or curated datasets to gain better domain knowledge (Kaplan et al., 2020; Chowdhery et al., 2022). In addition, some studies (Wang et al., 2022b; Wei et al., 2021; Sanh et al., 2021; Ouyang et al., 2022) have fine-tuned the pre-trained model alongside the prompts to enhance the mutual reinforcement between the model and the prompts. Moreover, Wang et al. (2022a) proposes to use self-consistency to improve the generalization performance of the prompts. These methods often require complex fine-tuning and model add-ons, which can reduce their accessibility if we are constrained by training resources such as the size of the downstream dataset.

**Prompt-engineering** In the context of hard prompt tuning for natural language processing (NLP) tasks, *prompt-engineering* or *reprogramming* are often used to better condition the pre-trained model at inference time. Previous work has found that the prompt format can change the model's behavior, and researchers have proposed various formats to better suit the downstream setting of a narrow or diverse task nature, type, or the size of the pre-trained model (Wei et al., 2022; Jung et al., 2022; Arora et al., 2022). Other methods require users to manually craft prompts in predefined formats that convey task information to the models (Mishra et al., 2022). Despite their comprehensiveness in handling natural language tokens, it can be challenging and less than ideal to create distinctive and suitable formats for each inference example across numerous domains. Our work is motivated by (Arora et al., 2022; Zhang et al., 2022; Wang et al., 2022c) wherein we automatically adjust a set of prompting tokens to match the inference samples. This helps to generalize the guiding signal better

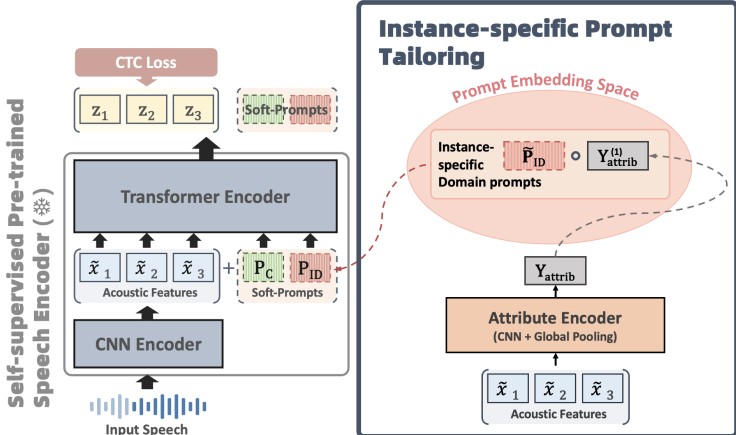

Figure 1: An overview of soft prompt tuning in speech recognition with the self-tailoring mechanism: In the standard configuration, the speech recognition system employs a universal set of **common prompts** $\mathbf{P_C}$, as typically seen in the vanilla setup. These prompts serve to capture general information relevant to the downstream ASR task. Alongside these, we introduce utterance-specific prompts that extract pertinent domain knowledge from the acoustic features and instill this information into the prompt vectors $\tilde{\mathbf{P}}_{\mathbf{ID}}$, which correspond to **individual prompts** $\mathbf{P_{ID}}$. The model learns to perform this self-tailoring process during fine-tuning that enables the prompts to offer more precise and customized instructions at the individual instance level.

to adapt to different domain inputs. Nonetheless, prompt engineering is not directly applicable to speech models, primarily due to the absence of naturally interpretable tokens like in NLP tasks. We overcome this challenge by using soft tokens with a non-trivial contribution from the proposed novel self-tailoring module.

**Prompt domain adaptation** In efforts to enhance domain adaptability, earlier work (Goswami et al., 2023) introduces a sequence of vectors derived from inferencing utterance using a term-frequency-based approach, aiming to capture domain-specific keywords. These vectors are employed alongside general soft prompts, and the model utilizes gating mechanisms to combine a general soft prompt and domain-specific tokens. This infuses the domain knowledge with the keyword vectors to improve adaptability. Furthermore, Guo et al. (2022) has applied domain adversarial neural network techniques with supplementary regularization terms to enhance the transferability and sample efficiency of prompt tuning. In contrast, it is difficult to determine keyword frames from speech utterances, especially when the timestamp is unknown to the model, which makes it less feasible for speech downstream tasks. Additionally, our method avoids complicating the loss functions for downstream task learning by using a much simpler yet effective module to achieve a similar goal.

## 2 PRELIMINARIES: PARAMETER EFFICIENT SOFT-PROMPT TUNING

In this section, we formally introduce prompt tuning, a parameter-efficient tuning approach in the context of automatic speech recognition (ASR). We also present prompt gating, an improved variant of prompt tuning that exhibits SOTA performance in ASR under the motivation of PET. This work evaluates on both prompting frameworks to demonstrate the efficacy of the proposed self-tailoring mechanism across existing strong networks for more informed and advanced insights.

**Prompt Tuning (PT)** Given a set of training data $(X, Y)$, where $x \in X$ represents input speech utterances, and $y \in Y$ corresponds to their text transcriptions, our goal is to leverage a self-supervised pre-trained speech encoder to efficiently learn ASR with a minimal number of trainable parameters. To achieve this, we define a set of learnable prompt vectors, $\mathbf{P} \in \mathbb{R}^{n_p \times d}$, which consists of $n_p$ tokens with the same dimension size of $d$ as the latent transformer embeddings $\tilde{x} \in \mathbb{R}^{T \times d}$ of the utterance $x$. Here, $T$ denotes the sequential frames of $\tilde{x}$ after being processed by the acoustic convolutional encoder. During the feed-forward process, we prepend the soft-prompts to $\tilde{x}$, creating the augmented input matrix $\tilde{x}_p := [\tilde{x}, \mathbf{P}] \in \mathbb{R}^{(T+n_p) \times d}$. This newly formed matrix, $\tilde{x}_p$, now becomes

the latent input to the transformer blocks. Subsequently, PT optimizes the following loss function with respect to $\mathbf{P}$, while maintaining the pre-trained model parameters $\Theta$ in a frozen state.

$$\mathcal{L}_{PT} = -\sum_i \log \mathbb{P}(y_i \mid x_i; \Theta, \mathbf{P}) \tag{1}$$

In the course of model training, the soft prompts are updated to provide more effective instructions to modify the latent speech representation in an additive manner through the self-attention module.

**Prompt Gating (PG)** Despite the success of this approach, studies (Huang et al., 2023; Qian et al., 2022; Gu et al., 2022b) find that the vanilla prompt tuning model experiences challenges in effectively utilizing instructional soft-prompt vectors for complex multi-aspect inputs, such as speech. This occurs when the model learns multiple distinct functional roles of soft prompts (Ng et al., 2023b) (e.g., contextual-handling prompts, noise-handling prompts, prosody-handling prompts) to process the speech utterance for the downstream task, but struggles to allocate appropriate weights to these functional signals, leading to attribute degeneration. To overcome the problem, Ng et al. (2023a) proposes a supplementary gating module that learns to regulate the flow of instructions from the prompts at both the token-level and channel-dimensional-level, to better control the prompts for fine-tuning the latent representations. This has been reported to be effective in generalizing ASR in both clean and noisy domains, resulting in significant performance gains from the proposed framework. We present more rigorous computational details in Appendix A.1.

## 3 INSTANCE-SPECIFIC PROMPT TAILORING

While prompt tuning has made notable progress in automatic speech recognition, the existing prompting framework exhibits an inflexible *"one-size-fits-all"* approach. Typically, it learns a common set of soft prompts for all inference samples, without taking into account the utterance-specific information for individual samples. This utterance-specific information could provide targeted instructions, enabling the model to better process necessary environmental settings, such as the type and level of background noise corruption and speaker accent. Inspired by the concept of hard prompt engineering, where an optimal prompt format is customized for each sample in NLP tasks, we are motivated to develop a similar computational module that applies specifically to soft prompts in speech recognition. Most importantly, we want to achieve this with as few trainable parameters as possible, in order to maintain the objective of parameter-efficient tuning.

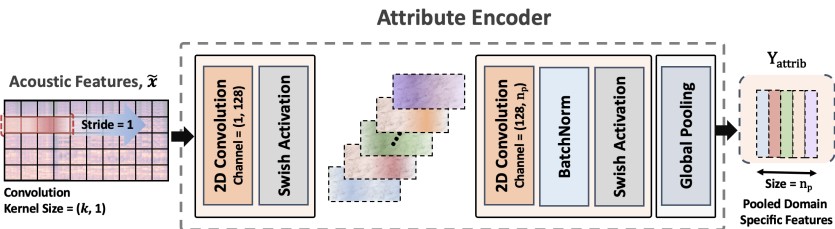

Figure 2: An illustration of the attribute encoder, designed to extract global utterance-specific information from intermediate acoustic features. In the default configuration, we employ two non-linear convolutional layers with a kernel size of 15 and a stride of 1 to enhance feature depth to the size of $n_p$, aligning it with the number of soft prompt tokens. Subsequently, global average pooling is applied to generate vector representations of the acoustic attributes.

**Utterance-Specific Prompt** To acquire a set of utterance-specific prompts, we begin with randomly initialized soft prompt vectors represented by $\tilde{\mathbf{P}}_{\mathbf{ID}}$. Then, we introduce a self-tailoring module that takes the intermediate acoustic features $\tilde{x}$ produced by the convolutional encoder of the self-supervised pre-trained speech model (i.e., analogous to the Fourier transform for timbral features) to generate relevant acoustic domain features through the design attribute encoder, as illustrated in Figure 2. The soft prompts, $\tilde{\mathbf{P}}_{\mathbf{ID}}$, are enriched with output attribute embeddings features, $\mathbf{Y}_{\mathbf{attrib}}$, through a parameterized decomposition method inspired by (Li et al., 2018; Wen et al., 2019; Aghajanyan et al., 2021; Wang et al., 2022c), where the resultant prompts learn to infuse information from the two representations. As such, this online tailoring of prompts to such individual level of more global timbral features helps the prompts provide more precise and targeted instructions for individual utterances, which improves the generalization of the downstream task.

Formally, we obtain $\tilde{\mathbf{Y}} \in \mathbb{R}^{n_p \times T \times d}$ through two non-linear convolutional layers, defined as $\tilde{\mathbf{Y}} = \sigma(\tilde{x}(t) * h(t))$, where $h(t)$ represents the kernel function of the convolutional layer, and $\sigma$ denotes the Swish activation function. In this process, we increase the depth of its feature map by $n_p$ to align with the number of soft prompt tokens. Then a global average pooling layer, $\texttt{pooling}(\tilde{\mathbf{Y}})$, is applied to obtain matrix $\mathbf{Y_{attrib}} \in \mathbb{R}^{n_p \times d}$, to encode the global utterance-specific timbral information from $\tilde{x}$ that will be helpful for ASR. We aim to instill this relevant utterance information into the soft prompts by decomposing matrix $\mathbf{Y_{attrib}}$ into $\tilde{\mathbf{P}}_{\mathbf{ID}}$. However, before the decomposition, we project $\mathbf{Y_{attrib}}$ into the embedding space of $\tilde{\mathbf{P}}_{\mathbf{ID}}$ by doing $\mathbf{Y_{attrib \rightarrow \tilde{P}}} = \mathbf{u_p u_p^\mathsf{T} Y_{attrib}}$, where $u_p$ is the left singular vectors of matrix $\tilde{\mathbf{P}}_{\mathbf{ID}}$ from its SVD, to ensure that they are from the same embedding space. Subsequently, the parameterization of the utterance-specific prompts is given by

$$\mathbf{P_{ID}} = \tilde{\mathbf{P}}_{\mathbf{ID}} \circ \mathbf{Y}^{(1)}_{\mathbf{attrib} \rightarrow \tilde{\mathbf{P}}} = \tilde{\mathbf{P}}_{\mathbf{ID}} \circ (u_Y^{(1)} v_Y^{(1)\mathsf{T}}) \,, \tag{2}$$

where $\circ$ represents the Hadamard product between two matrices, $u_Y^{(1)} \in \mathbb{R}^{n_p}$ and $v_Y^{(1)} \in \mathbb{R}^d$ refers to a rank one matrix of the SVD of $\mathbf{Y_{attrib \rightarrow \tilde{P}}}$. This design incurs a small incremental increase in trainable parameters, and the low-rank subspace approach (Aghajanyan et al., 2021) is reported to reduce the dependence on larger data size for optimization, making it suitable for low-resource setting. Consequently, it allows the "slow weights", $\tilde{\mathbf{P}}_{\mathbf{ID}}$, to capture general information across the downstream ASR task, while "fast weights", $\mathbf{Y_{attrib}}$, focus on acquiring relevant domain knowledge from the inferencing utterance within a low-rank subspace.

**Prompt Masking**   To improve the general quality of the prompt tokens used to fine-tune the speech representations, we are inspired by (He et al., 2022; Baevski et al., 2022; Park et al., 2019) to apply a simple masking strategy to the prompts to regularize the tokens, resulting in more robust performance. In particular, our approach involves two sub-processes:

1. **Prompt Dropping**: We randomly remove less than one-fifth of the prompt tokens from the defined number of prompts in the set (atypical to masking the token frames).
2. **Prompt Channel Masking**: We mask the channel dimensions of the remaining prompt tokens in a consecutive manner with no more than 20% of the original embedding size.

This simple approach appears to be effective in improving generalization, as it forces the prompts to optimize themselves to provide more meaningful representations for each token and across the channel dimension in the event when some of the prompts are missing or incomplete. Additionally, it helps to prevent the model from overfitting to specific patterns in the prompts on the training data.

**Redundancy Reduction Loss**   Prior studies (Ng et al., 2023a;b) indicate that the soft prompts learned from the vanilla prompt tuning framework for ASR contain many highly correlated tokens. This leads to unnecessary inefficiency, as we typically have multiple prompt tokens. Ideally, we want each token to provide distinctive information, reducing redundancy in the function of the prompts.

Therefore, we compute the correlation matrix among the prompt tokens and minimize it to an identity matrix (Zbontar et al., 2021; Peyser et al., 2022; Chang et al., 2023a), as an auxiliary loss. This reduces the correlation between the prompt tokens, as the off-diagonal elements of the prompt correlation matrix $C_{ij}$ measure the relatedness between each token and others. Computationally, we can ignore the diagonal term of the correlation matrix because the correlation of a variable with itself is always 1. We define the redundancy reduction loss function as

$$\mathcal{L}_{\mathrm{RR}} \triangleq \sum_i \sum_{j \neq i} C_{ij}^2 \tag{3}$$

Finally, the total loss function for training the proposed prompt-tuning framework is represented by

$$\mathcal{L}_{\mathrm{Total}} = \mathcal{L}_{\mathrm{CTC}} + \lambda \mathcal{L}_{\mathrm{RR}} \tag{4}$$

where $\mathcal{L}_{\mathrm{CTC}}$ computes the Connectionist Temporal Classification (CTC) loss for ASR fine-tuning and $\lambda$ is a weighting parameter that determines the emphasis on the redundancy in prompt tokens.

## 4   EXPERIMENTS

In this section, we outline our experimental design for self-tailoring prompt tuning and present the results with insights. We compare self-tailoring prompts to the vanilla prompt tuning method and

other commonly used Parameter-Efficient Tuning approaches, providing a comprehensive assessment of their performance. Our evaluation encompasses various open-world scenarios, including clean speech recognition, simulated noisy speech recognition, and testing on real-world out-of-domain noisy speech to ensure real deployment relevance.

**Datasets**  The downstream ASR fine-tuning is conducted mainly using the official 100 hours of LibriSpeech Corpus (Panayotov et al., 2015). Furthermore, the training dataset for noisy audio is synthesized by mixing the aforementioned set with FreeSound noise (Font et al., 2013) at 0 to 20dB. The noise dataset comprises both stationary (Type A) and non-stationary (Type B) noises, including Car, Metro, Traffic for type A and Babble Airport/Station, Cafe, AC/Vacuum noises, for type B, respectively. Each noise type has 10 audio streams for training and 8 for testing, totaling approximately 2 hours of noise data. During testing, we use the same pre-mixed test set from Prasad et al. (2021), which we reference as *Test-Noisy* in our experimental results. This set randomly selects 120 sub-files from test-clean LibriSpeech and corrupts them with the other 8 partition noise streams at 0 to 20 dB, resulting in 4200 instances. The noise data and the synthesized noisy test set can be downloaded from this link[1]. Furthermore, we incorporate real-world noisy data from the CHiME4 challenge (Vincent et al., 2017) to assess the model's noise robustness in out-of-domain scenarios.

**Models**  Following the prompt tuning approach (Chen et al., 2023; Ng et al., 2023a), we present our experimental results using WavLM Base+ (Chen et al., 2022), the latest self-supervised pre-trained speech encoder with 94.7M parameters. This model underwent pre-training on 94k hours of diverse data sources, including LibriLight (Kahn et al., 2020), VoxPopuli (Wang et al., 2021), and GigaSpeech (Chen et al., 2021), with additional background noise augmentation, resulting in robustness across various audio backgrounds, speakers, and content types. In our prompt tuning framework, we initialize two sets of prompts: common sharing prompts, which adhere to the standard setup and capture general downstream task information, and self-tailoring prompts, which adapt prompt vectors to each individual utterance for customized, instance-specific behavior. Each set comprises an equal number of tokens, and the add-up of both matches the number of tokens in the vanilla approach for fair comparison. As our default configuration, we initialize a total of 100 tokens for prompt tuning and 10 tokens for prompt gating. The decision on the number of prompts was determined based on achieving the optimal balance between performance and the number of trainable parameters employed. Besides, we report the experiments with the popular HuBERT encoder in Appendix A.2 to explore the generalization performance across pre-training models, providing a more comprehensive investigation of our method.

**Baselines**  We evaluate ASR performance through a comparison with full-model fine-tuning and several commonly used Parameter-Efficient Tuning methods, consisting of different variants of vanilla Prompt Tuning Lester et al. (2021); Ng et al. (2023a), Adapter Tuning Houlsby et al. (2019), LoRA Hu et al. (2021), and a 'Frozen' model scenario where we freeze the entire upstream WavLM Base+ encoder while training only the downstream predictor module.

**Implementation Details**  The downstream ASR training strictly follows the configurations of the standard setup outlined in the FairSeq (Ott et al., 2019) training script. Generally, we freeze the encoder parameters and exclusively update the Parameter-Efficient Tuning module to perform the efficient training. The down-sampling projection rate for Adapter Tuning is set at 4 (dim=192) and uses only one adapter module at the feed-forward layer, while the low-rank decomposition factor for LoRA is set to 16. The hyperparameter $\lambda$ for $\mathcal{L}_{\text{RR}}$ is set to 0.05 in our work. We perform a grid search for the learning rate (LR) within the [2e-5, 3e-4] range, with 1e-4 LR yielding optimal results on LibriSpeech's Dev-Clean set for the efficient tuning methods. Lastly, we apply an RNN decoder network, as described by Chen et al. (2023), to decode the features derived from the encoder output, with its architectural design and the training procedures consistent with SUPERB (Yang et al., 2021) for all efficient tuning methods.

## 4.1 RESULTS

Table 1 presents the performance of different PET methods for speech recognition on the official LibriSpeech evaluation sets. We measure their accuracy using Word Error Rate (WER), with lower values indicating a more accurate recognition system. The models in this table are trained on the original 100-hour LibriSpeech corpus, and tested without the use of a language model, allowing

---

[1] https://github.com/archiki/Robust-E2E-ASR

Table 1: Results on the original official LibriSpeech evaluation set with 100 hours of training data. The table shows the word error rate, WER (%) (↓) of the ASR system on clean speech recognition without a language model.

| Models | Params (M) | Dev | | Test | |
|---|---|---|---|---|---|
| | | Clean | Other | Clean | Other |
| WavLM+ (Chen et al., 2022) | 94.70 | 4.67 | 10.22 | 4.71 | 10.27 |
| WavLM+ (Frozen) (Chen et al., 2022) | 0 | 5.03 | 11.91 | 5.17 | 11.97 |
| Adapter (Houlsby et al., 2019) | 3.54 | 4.41 | 9.96 | 4.56 | 9.88 |
| LoRA (Hu et al., 2021) | 0.59 | 5.30 | 12.15 | 5.49 | 12.19 |
| Prompt Tuning (Lester et al., 2021) | 0.08 | 4.90 | 11.88 | 5.05 | 11.86 |
| Self-Tailoring PT (Ours) | 0.34 | **4.66** | **11.39** | **4.80** | **11.28** |
| Prompt Gating (Ng et al., 2023a) | 0.47 | 4.30 | 9.97 | 4.55 | 9.85 |
| Self-Tailoring PG (Ours) | 0.65 | **4.06** | **9.48** | **4.31** | **9.36** |

us to assess the effectiveness of the PET architectures independently. Note that the results for the test-clean set are comparable to the SUBERB benchmark for ASR. From the table, we observe improved performance for the frozen WavLM Base+ model, achieving a WER of 5.17, in contrast to the reported score of 5.59 Chen et al. (2022). This suggests that our table benchmark is highly competitive, and we strive to optimize it to its optimal state. When comparing existing PET methods to full-model fine-tuning, we observe that nearly all PET approaches with highly efficient designs (i.e., having $< 1\%$ of the trainable parameters of the full model) struggle to match the performance of full-model fine-tuning. This challenge likely arises from the difficulty of maximizing the network capacity of these compact computational modules to compete with the large transformer encoder (94.7M). However, we show that our self-tailoring module overcomes this challenge and achieves improved performance, with a relative gain of $(4.9 - 5.3)\%$ on the testing set compared to the variants of the vanilla prompt tuning method, despite its small incremental number of trainable parameters. We highlight that simply increasing the trainable parameters of vanilla prompt tuning by increasing the number of tokens to the same level as self-tailoring prompts does not improve performance and can in fact cause overfitting, resulting in a drop in performance. Notably, self-tailoring prompt gating outperforms the full fine-tuning model by a relative gain of $(8.5 - 8.9)\%$ on the testing set, highlighting the effectiveness of formatting vanilla soft prompts into utterance-specific prompts, allowing the model to optimize more effectively for the ASR task.

Table 2: Results on the synthesized noisy LibriSpeech (FreeSound) with 100 hours of in-domain noise data. The table shows the word error rate, WER (%) (↓) of the ASR system on noisy speech recognition at SNRs of (0 - 20)dB without a language model.

| Models | Params (M) | Non-Stationary (Type-B) Noise | | | | Stationary (Type-A) Noise | | | Avg. (Noisy) |
|---|---|---|---|---|---|---|---|---|---|
| | | Babble | Airport/ Station | AC/ Vacuum | Cafe | Traffic | Metro | Car | |
| WavLM+ | 94.70 | 14.67 | 10.66 | 10.83 | 8.56 | 8.63 | 8.60 | 5.94 | 9.70 |
| WavLM+ (Frozen) | 0 | 25.26 | 17.55 | 15.96 | 12.35 | 12.09 | 11.72 | 7.24 | 14.60 |
| Adapter | 3.54 | 16.16 | 11.95 | 12.02 | 9.30 | 9.29 | 9.04 | 6.21 | 10.57 |
| LoRA | 0.59 | 35.68 | 26.08 | 21.68 | 15.61 | 14.34 | 14.06 | 7.94 | 19.34 |
| Prompt Tuning | 0.08 | 18.81 | 13.55 | 13.24 | 10.27 | 10.43 | 9.83 | 6.43 | 11.79 |
| Self-Tailoring PT (Ours) | 0.34 | **18.09** | **12.84** | **12.65** | **9.74** | **9.89** | **9.25** | **6.12** | **11.23** |
| Prompt Gating | 0.47 | 15.92 | 11.51 | 11.19 | 9.05 | 9.18 | 8.65 | 5.67 | 10.17 |
| Self-Tailoring PG (Ours) | 0.65 | **14.88** | **10.83** | **10.58** | **8.52** | **8.67** | **8.37** | **5.44** | **9.61** |

Table 2 displays the generalization performance on noisy (in-domain) speech recognition, where the models are trained on 100 hours of LibriSpeech corrupted with 0 to 20dB of FreeSound noises. We observe a significant widening performance gap between the full fine-tuning model and the rest

of the PET methods. While prompt gating remains the best-performing candidate among the PET models, it is clear that these efficient frameworks are incomparable to full model fine-tuning in handling background noises, even though WavLM Base+ is pre-trained to be robust to noisy audio. This suggests that the model may still need adjustments in its full architecture to adapt optimally to the target noise for improved noise robustness. However, our self-tailoring prompts have consistently improved the recognition performance by a relative gain of $(4.7 - 5.5)\%$. Surprisingly, self-tailoring prompt gating championed full-model fine-tuning, even though it learns in a more challenging environment with only 0.7% of full model trainable parameters. This finding supports the idea that utterance-specific prompts contribute to better adaptation to the environment, thereby enhancing overall model generalization.

To further investigate the generalization performance on out-of-domain noise setting, we evaluate the trained models from Table 2 on real-world recorded noisy speech from the CHiME4 challenge. Besides, evaluating speech corpus recorded in real-world noisy environments provides us with a better understanding of how the models perform on more realistic audio that is relevant to the real world.

Table 3 shows that the performance gap between inefficient full fine-tuning and the reported PET methods remains significant. We note that LoRA performs even worse than the frozen base encoder, both in-domain and out-of-domain. Empirically, we found that the low-rank assumption for transformer weights may not be applicable to ASR, as it requires more than learning a style transfer, like in NLP. Instead, it requires mapping speech representations to natural text, which may require more effort and information, resulting in a much higher rank term. Then, while prompt tuning variants improve the performance of the frozen WavLM Base+ model, closing the gap with the full-model fine-tuned WavLM Base+, they generally underperform the full model. However, self-tailoring prompts consistently improve out-of-domain performance by $(4.4 - 6.6)\%$, beating even full-model tuning with self-tailoring prompt gating.

Table 3: Results on the out-of-domain evaluation of CHiME4 real 1-channel testing set.

| Models | Params (M) | Dev | Eval |
|---|---|---|---|
| WavLM+ | 94.70 | 15.86 | 19.42 |
| WavLM+ (Frozen) | 0 | 20.37 | 26.35 |
| Adapter | 3.54 | 16.24 | 19.85 |
| LoRA | 0.59 | 24.09 | 32.12 |
| Prompt Tuning | 0.08 | 18.32 | 22.52 |
| Self-Tailoring PT (Ours) | 0.34 | **17.51** | **21.41** |
| Prompt Gating | 0.47 | 16.17 | 19.34 |
| Self-Tailoring PG (Ours) | 0.65 | **15.11** | **18.37** |

## 4.2 ABLATION STUDIES

Table 4: Ablation results on the sub-components of the proposed method. The reported scores are based on the average WER obtained from Test-Noisy (from Table 2) and CHiME4 real eval corpus.

| Models | Self-Tailoring PT | | Self-Tailoring PG | |
|---|---|---|---|---|
| | Test-Noisy | CHiME4-Eval (Real) | Test-Noisy | CHiME4-Eval (Real) |
| Base (Default Setup) | 11.23 | 21.41 | 9.61 | 18.31 |
| ➥ (-) Prompt Masking | 11.42 | 21.92 | 9.75 | 18.73 |
| ➥ (-) Redundancy Reduction Loss | 11.48 | 22.08 | 9.81 | 18.91 |
| ➥ (-) Prompt Tailoring | 11.58 | 22.39 | 9.92 | 19.18 |

To assess the effectiveness of each sub-component of the propose method, we evaluate the model's error rate performance in Table 4. This evaluation involves removing one sub-component at a time and measuring the performance. Based on the increase in error rates, we conclude that prompt tailoring has the biggest impact, followed by the redundancy reduction loss, and then prompt regularization. Specifically, removing prompt tailoring causes a relative increase in WER by as much as 4.8%. Furthermore, we observe that encouraging the functionality of the prompts to be non-sharing seems to have a substantial influence on the model's performance. This diversifies the prompts, enhancing the effectiveness of the prompt signals in fine-tuning the representations.

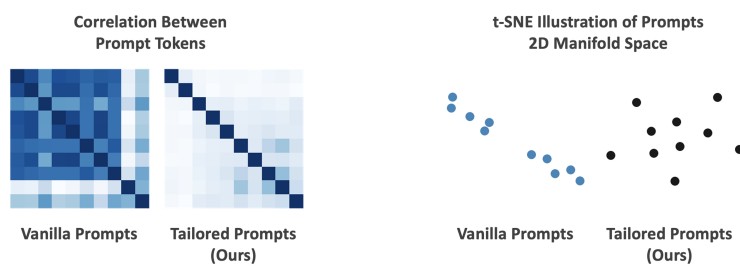

Figure 3: Figure visualization of prompt tokens from vanilla and self-tailoring prompt gating, using their correlation matrix and 2D t-SNE representations.

Similarly, in Figure 3, we present a visualization plot of the prompt tokens belonging to the self-tailoring prompt gating in the 2D t-SNE manifold space and its correlation matrix to further support the motivation of using the redundancy reduction loss. The correlation plot takes absolute values because we are only interested in whether the prompts are distinctive from each other, as indicated by low correlation values (lighter shades of blue, with white corresponding to 0). Then, we see that the vanilla prompts are highly inefficient and many of them are correlated. The t-SNE plot shows two clusters of similar data points within each cluster, which may imply inefficiencies in handling various aspects of speech for ASR, such as content and noise. However, with the auxiliary loss, the prompts become more informative and are well-spread in the t-SNE plot.

Table 5: Results of prompt tuning variants on the Test-Noisy (in-domain) and CHiME4 (out-of-domain) real-world noisy speech evaluation sets over different training corpus sizes.

| Models | Test-Noisy | | | CHiME4-Eval (Real) | | |
|---|---|---|---|---|---|---|
| | 10h | 100h | 360h | 10h | 100h | 360h |
| Prompt Tuning | 20.45 | 11.79 | 9.89 | 37.37 | 22.52 | 20.05 |
| Self-Tailoring PT (Ours) | **19.84** | **11.23** | **9.51** | **36.00** | **21.41** | **19.26** |
| Prompt Gating | 16.79 | 10.17 | 8.45 | 32.89 | 19.34 | 17.16 |
| Self-Tailoring PG (Ours) | **16.23** | **9.61** | **8.07** | **31.62** | **18.37** | **16.31** |

Lastly, we evaluate the model's behavior with different training sizes, training prompt tuning models with 10, 100, and 360 hours of synthesized noisy LibriSpeech (following Table 2). We then test the model on the in-domain Test-Noisy set and the real-world noisy speech from the out-of-domain CHiME4 challenge. Table 5 shows that the proposed self-tailoring prompt generalizes well over different training sizes, with a relative gain of $(2.7 - 5)\%$. However, when the data is extremely scarce (e.g., 10 hours), the advantage of our method is less prominent compared to other instances when the training size is greater than 100 hours. We believe that our attribute encoder, which contributes to the utterance-specific prompts, is still limited by its requirement for sufficient data samples to better generalize its function. We plan to replace this module with a pre-trained network in future work.

## 5 CONCLUSION

In conclusion, we introduce self-tailoring prompts, which adaptively modify prompt tokens by incorporating relevant utterance-specific information. This modification of conventional standard-template soft prompts results in more targeted signals for enhancing representation tuning. We find that our approach is highly effective in improving downstream generalization and consistently achieves improved performance across a variety of audio environmental domain scenarios. Besides, the proposed regularization techniques have been demonstrated to successfully enhance the quality of prompt tokens. Remarkably, our method surpasses the performance of the full fine-tuning model while utilizing as little as 0.7% of its trainable weights.

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

# A APPENDIX

## A.1 PROMPT GATING: COMPUTATION ILLUSTRATION

Human speech is a multi-faceted signal (e.g., content, speaker prosody, and background noise distortion), and these factors appear to be important information to process to assist a recognition system in generalizing well to the domain environment. From Figure 3, we can clearly see two clusters in the t-SNE projection of the 2D manifold space for the vanilla method of soft prompts. These clusters may correspond to distinct sets of prompting information that handles different aspects of speech, as Ng et al. (2023b) has shown in previous studies. Although prompt tuning provides very efficient and effective representation tuning for some tasks, it may face challenges in handling complex multi-aspect features and allocating appropriate weights to these factors. Hence, a gating module is designed for the prompts, where an external module can better regulate the information flow from the prompts to handle the frames of the utterance and decide on the optimal use of the sets of information for speech recognition.

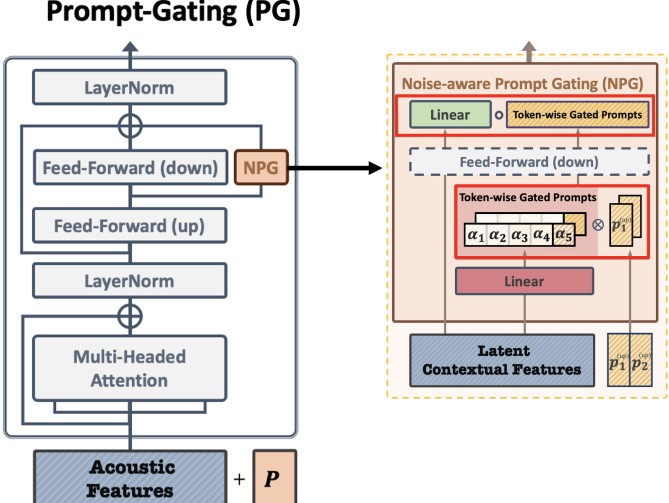

Figure 4: Figure illustration of prompt gating. $P$ refers to the soft-prompt tokens used in tuning representations for the downstream ASR.

To do this, the gating module uses two functional layers that take into account the latent utterance features to gate the token-level and channel-wise features of the prompt tokens. It first assesses each frame of the speech utterance to predict the gating weights of a single value coefficient for each prompt token (token-level) of the forwarding sequence. Then, another linear layer takes into account the acoustic features to predict a (channel-wise) gating mask for the prompt channel features, which is then applied to the prompt sequence. Consequently, the optimized gated prompt output is obtained and additively added to the latent representations after the feed-forward layer of the transformer block. Formally, the computation is given by

$$P_t' = \delta(W_t.\tilde{x}') \otimes P' \tag{5}$$

$$P_t = W_{\text{FFN}}^{(\text{down})}.P_t' \tag{6}$$

$$P_{c \to t} = \delta(W_c.\tilde{x}) \circ P_t \tag{7}$$

where $P' \in \mathbb{R}^{n_p \times 3072}$ takes the channel dimensional size of the feed-forward up layer in the transformer block (i.e., dim = 3072 in our case), $P_t'$ denotes the gated token-level prompts of the sequence and $P_t \in \mathbb{R}^{n_p \times 768}$ is the down-sampled $P_t'$ that re-uses the down-sampling layer of the feed-forward block. This computation takes place at a higher dimension to exploit higher-granularity information for better prediction of token-level gating weights. $P_{c \to t} \in \mathbb{R}^{n_p \times 768}$ refers to the channel-wise gated prompts applying on $P_t$. However, to simplify token-level gating, we perform all computations at the base dimension of 768. This also reduces the trainable parameters needed for the gating module to become more efficient, albeit at a slight cost to performance accuracy.

A.2    AN EVALUATION ON HUBERT ENCODER

In this section, we repeat the evaluation conducted previously with the same training configurations (i.e., on **clean speech, noisy in-domain speech and out-of-domain real-world noisy speech** environments), but instead of utilizing the WavLM Base+ as the self-supervised pre-trained speech encoder, we substitute it with another widely-used option, the HuBERT Base speech encoder, to analyze performance. HuBERT Base is pre-trained on a smaller dataset of 960 hours, primarily consisting of clean corpus data. Our aim is to determine whether we can maintain the advantages provided by the proposed self-tailoring prompts and assess the model's robustness to noise in this different pre-training setting.

Table 6: Results on the original official LibriSpeech evaluation set with 100 hours of training data. The table shows the word error rate, WER (%) (↓) of the ASR system on clean speech recognition without a language model.

| Models | Params (M) | Dev | | Test | |
|---|---|---|---|---|---|
| | | Clean | Other | Clean | Other |
| HuBERT | 94.70 | 5.87 | 13.13 | 5.92 | 12.95 |
| HuBERT (Frozen) | 0 | 6.21 | 17.11 | 6.52 | 17.29 |
| Adapter | 3.54 | 4.97 | 12.39 | 5.18 | 12.13 |
| LoRA | 0.59 | 6.13 | 16.88 | 6.48 | 17.15 |
| Prompt Tuning | 0.08 | 5.87 | 15.85 | 6.20 | 15.94 |
| Self-Tailoring PT | 0.34 | **5.53** | **15.21** | **5.82** | **15.24** |
| Prompt Gating | 0.47 | 5.10 | 12.26 | 5.34 | 12.13 |
| Self-Tailoring PG | 0.65 | **4.83** | **11.74** | **5.07** | **11.62** |

From Table 6, we can observe that most efficient tuning methods still appear to be inferior compared to the non-efficient tuning of the full model. However, the performance gap is more noticeable when using the HuBERT Base encoder, possibly due to the relatively weaker generalization of speech representations from the smaller pre-training dataset and the absence of the relative positional bias found in the architecture of WavLM. Despite these challenges, we note that the self-tailoring prompt variants have achieved a relative gain of (4.2 – 6.1)% on the LibriSpeech test set. Furthermore, self-tailoring prompt gating has also outperformed the full fine-tuning model by as much as 14.4% relative improvement, despite its extreme efficiency in fine-tuning.

Table 7: Results on the synthesized noisy LibriSpeech (FreeSound) with 100 hours of in-domain noise data. The table shows the word error rate, WER (%) (↓) of the ASR system on noisy speech recognition at SNRs of (0 - 20)dB without a language model.

| Models | Params (M) | Non-Stationary (Type-B) Noise | | | | Stationary (Type-A) Noise | | | Avg. (Noisy) |
|---|---|---|---|---|---|---|---|---|---|
| | | Babble | Airport/ Station | AC/ Vacuum | Cafe | Traffic | Metro | Car | |
| HuBERT | 94.70 | 25.87 | 19.93 | 17.78 | 14.17 | 13.58 | 13.22 | 8.83 | 16.20 |
| HuBERT (Frozen) | 0 | 49.98 | 39.32 | 31.98 | 26.49 | 23.15 | 20.83 | 11.53 | 29.04 |
| Adapter | 3.54 | 32.69 | 24.71 | 23.57 | 16.58 | 15.19 | 15.87 | 8.17 | 19.54 |
| LoRA | 0.59 | 54.43 | 42.94 | 36.88 | 28.87 | 24.51 | 23.17 | 11.14 | 31.71 |
| Prompt Tuning | 0.08 | 44.88 | 34.74 | 30.32 | 23.66 | 20.42 | 19.69 | 9.95 | 26.24 |
| Self-Tailoring PT | 0.34 | **41.22** | **31.50** | **29.04** | **21.77** | **19.14** | **18.41** | **9.18** | **24.32** |
| Prompt Gating | 0.47 | 33.71 | 25.19 | 22.89 | 15.98 | 15.18 | 15.34 | 8.09 | 19.48 |
| Self-Tailoring PG | 0.65 | **32.14** | **23.23** | **21.74** | **14.63** | **14.45** | **14.56** | **7.49** | **18.32** |

Table 7 presents the performance on in-domain noisy speech. Once again, we can observe consistent performance gains with the proposed self-tailoring prompts. However, interestingly, we were unable to surpass the non-efficient tuning of the full model in this altered environment of noisy speech. This

could likely be attributed to the fact that the trainable parameters and network complexity of the efficient models might not be sufficient to learn noise handling from scratch, especially considering that the pre-trained models are not inherently robust to noise.

Table 8: Results on the out-of-domain evaluation of CHiME4 real 1-channel testing set.

| Models | Params (M) | Dev | Eval |
|---|---|---|---|
| HuBERT | 94.70 | 25.82 | 31.22 |
| HuBERT (Frozen) | 0 | 38.36 | 48.56 |
| Adapter | 3.54 | 29.60 | 38.43 |
| LoRA | 0.59 | 41.43 | 54.08 |
| Prompt Tuning | 0.08 | 34.36 | 46.90 |
| Self-Tailoring PT (Ours) | 0.34 | **32.58** | **44.82** |
| Prompt Gating | 0.47 | 29.39 | 38.86 |
| Self-Tailoring PG (Ours) | 0.65 | **28.04** | **37.25** |

Finally, we assess the models generalization performance in an out-of-domain setting, where we evaluate the system using real-world noisy speech from CHiME4. In comparison to the performance achieved with WavLM Base+, HuBERT experiences a notable increase in error rate. This is likely due to the pre-trained model's lack of noise robustness. However, our methods still provide an advantage in enhancing its recognition capability. This supports the notion that utterance-specific prompts play a crucial role in improving adaptation to diverse environments, thereby enhancing overall model generalization.

