# OpenReview forum: "SELF-TAILORING PROMPTS FOR PARAMETER EFFICIENT TUNING SPEECH RECOGNITION"
_ICLR.cc/2024/Conference — ICLR 2024 Conference Withdrawn Submission_

### Official Review · Reviewer_A9xv · 2023-10-17

**Soundness:** 2 fair
**Presentation:** 3 good
**Contribution:** 3 good
**Rating:** 3
**Confidence:** 4

**Summary:**

This paper proposes an utterance-wise soft prompt learning for ASR efficient finetuning. The key idea is to have a set of learnable embeddings, which are generated by an attribute encoder, encoding the utterance specific environmental information. The learned embeddings can then be used as the utterance-level soft prompts. During ASR training and inference time, these soft prompts are appended/prepended as the transformer input.

**Strengths:**

* The idea of learning utterance-specific embedding has been shown success in the speech synthesis area. It is interesting to see that this can improve ASR performance as well.
* The organization, presentation, and references are satisfactory.

**Weaknesses:**

* The paper is motivated by “large speech recognition models”. However, the experiments are conducted on a  94.7M model, which is similar to other ASR models (ConformerL) that can be efficiently trained from scratch. Therefore, “PET” is less of a need for such models. It would be good to experiment with larger model sizes (billions). Otherwise, I would suggest restricting the scope to self-supervised ASR only.
* The size/cost of the attribute encoder is not described in the paper. This encoder is needed during both training and inference time. Therefore, the current comparison is not fair without considering the encoder. Would the baselines have a better WER when adding the number of parameters of the attribute encoder? Extra ablation studies are needed to verify this.
* It would be interesting to see the WERs with additional LM rescoring, or applying the same technique to the models with internal LM modeling, e.g., RNN-T. Those models are more robust, and it would be interesting to see if utterance-wise soft prompts can further improve their robustness.

**Questions:**

See weaknesses.

---

### Official Review · Reviewer_McqM · 2023-10-27

**Soundness:** 2 fair
**Presentation:** 3 good
**Contribution:** 1 poor
**Rating:** 3
**Confidence:** 4

**Summary:**

Authors address parameter efficient fine-tuning of large speech models for automatic speech recognition (ASR). They focus on soft-prompt tuning and propose a self-tailoring mechanism for soft prompts in ASR which is evaluated on several scenarios including clean and noisy acoustic environments. Self-tailoring means that individual prompts (tailored for each specific utterance/instance) are used in addition to the common prompts used in the vanilla setup. Experiments are made using WaveLM speech encoder and Librispeech-100h training set to validate the approach proposed.

**Strengths:**

-prompt tuning is not so common in speech processing and ASR

-the method proposed is an alternative to Adapter or LoRa parameter efficient fine tuning

**Weaknesses:**

-models fine tuned on librispeech100h training setting (read speech) only and performance are overall disappointing (see below)

-improvements of the proposed approach (with best model - prompt gating) over adapter fine tuning are very tiny (<0.5%) and the self-tailored prompt tuning seems to not work better than simple adapter fine tuning

-the problem addressed is kind of a niche (another parameter efficient fine tuning for ASR whereas Adapter are working well) and it is not even sure that - as speech models remain rather small (a few million parameters) -  parameter efficient fine tuning (instead of full model fine tuning) is really needed in ASR at the moment

**Questions:**

-what is captured by the attribute encoder ? some analysis on this would have been a plus

-the role of ‘Redundancy Reduction Loss’ part should be clarified

-why does LoRa do not work at all?

---

### Official Review · Reviewer_iM33 · 2023-10-30

**Soundness:** 4 excellent
**Presentation:** 4 excellent
**Contribution:** 4 excellent
**Rating:** 8
**Confidence:** 3

**Summary:**

This paper presents a self-tailoring prompting mechanism to address the limitations of existing prompting frameworks in speech recognition. The mechanism includes prompt masking regularization techniques and a redundancy reduction loss to enhance the quality of prompt tokens. It outperforms the full fine-tuning model using only a fraction of its trainable weights.

**Strengths:**

1. In this paper, the motivation of the self-tailoring mechanism is clear（incorporating utterance-specific information & improving generalization capability).

2. The introductions of Utterance-Specific Prompt and Redundancy Reduction Loss are novel.

3. The experimental part is very solid. Overall, the method proposed by authors achieves better results than LoRA and other Parameter-Efficient Tuning methods. And it outperforms the full fine-tuning model using only a fraction of its trainable weights.

**Weaknesses:**

1. Limited Noise Handling in Noisy Speech: in Table 7, the proposed method  is unable to surpass the non-efficient tuning of the full model in this altered environment of noisy speech.

2. Limited Prominence at Extremely Scarce Training Sizes: in Table 5, while the method generalizes well across different training sizes, its advantage becomes less prominent when the training size is extremely scarce.

**Questions:**

1. In the attribute encoder, why not use Attention to aggregate this global feature? The convolution layers are too simple.

---

### Official Review · Reviewer_QFFv · 2023-11-01

**Soundness:** 3 good
**Presentation:** 3 good
**Contribution:** 2 fair
**Rating:** 3
**Confidence:** 5

**Summary:**

This paper improves a technique called "prompt tailoring" imported from text LLM to speech SSL models for fine-tuning on speech recognition. The improvement comes from two novelties: 1. a "Self-tailoring mechanism" that extract from the acoustic features new prompt tokens containing other information than the one from the transformer embedding (directly into the prompt tokens) using a CNN. 2. a regularisation loss that tries to ensure that non-redundant information remains in the learned prompt after fine-tuning. The paper is clear and relatively well-written. The results also are convincing as better or equivalent performance compared to other techniques or full fine-tuning are reported on Librispeech (very clean use case) and CHIME-4 (very noisy use case).

**Strengths:**

The major strength of this paper is to apply something simple that sounds logical to do (see weaknesses) and obtain good performance with it.

**Originality**
To my knowledge, this is the first paper to perform speech embedding adaptation in the **context of prompt fine-tuning**. In this context, it is novel. Although, weaknesses apply (see bellow).

**Quality**
The paper is of sufficient quality to appear in a good speech conference. The two proposed solutions are replicable, and, despite not being well-grounded in the literature, based on techniques that are common in the speech domain -- i.e. this is solid.

**Clarity**
The paper is relatively clear. The two ideas are clearly explained and enough information is given to replicate them.

**Weaknesses:**

Two major issues arise with this work: the lack of connection to the speech literature and the minor impact of the findings.

The paper is written as an import of NLP technologies to the speech domain, like many others. We often see from such papers that they tend to bypass entirely the state-of-the-art from such domains. Again, the work is interesting and worth exploring, although we knew from the beginning that it would work - hence a limited impact. Indeed, the depicted techniques are basically an extension of feature and model adaptation - techniques that have been investigated since 1990 in the speech domain. Here, I am not saying that this particular technique exists as it is described in this paper, I am mentioning that we already knew that this concept would work - this paper gives us empirical proof. Hence, it is valuable, but I doubt that this is sufficient for ICLR. I would tend to say that such a work would have much more impact at Interspeech than ICLR. The paper, in its current form, does a good job of comparing with the NLP literature on prompting, but the thing is - this is speech recognition, and people have been trying to fine-tune ASR models for 20 years now (and SSL ones for 3-4 years), and we can't find a word about it in the document. "Prompt engineering" is a naming imported from NLP to speech, while we already had something to describe this idea - ASR adaptation.  ASR models have been adapted following dozens of ideas, including incorporating embedding vectors concatenated or merged with the hidden representation of the ASR model (as proposed in this work). These embedding could be frozen or learned alongside with the ASR fine-tuning, and also come from the input signal (typical speaker adaptation with x/i vectors).

**Questions:**

No question as the paper is pretty clear.
I will wait for the discussion period to engage with the authors and reviewers.

---

### Comment · Area_Chair_EAd4 · 2023-11-10
**reviewer-author discussions**

Dear All,

The reviewer-author discussion period will be from Nov. 10 to Nov. 22. For reviewers, please read the authors' responses and acknowledge it, respond to them early on in the discussion, and discuss points of disagreement. Thank you!

AC